# COVID-19 and the public response: Knowledge, attitude and practice of the public in mitigating the pandemic in Addis Ababa, Ethiopia

Zelalem Desalegn[1]*, Negussie Deyessa[2], Brhanu Teka[1], Welelta Shiferaw[3], Damen Hailemariam[2], Adamu Addissie[2], Abdulnasir Abagero[2], Mirgissa Kaba[2], Workeabeba Abebe[4], Berhanu Nega[5], Wondimu Ayele[2], Tewodros Haile[6], Yirgu Gebrehiwot[7], Wondwossen Amogne[6], Eva Johanna Kantelhardt[8], Tamrat Abebe[1]

1 Department of Microbiology, Immunology and Parasitology, School of Medicine, College of Health Sciences, Addis Ababa University, Addis Ababa, Ethiopia, 2 School of Public Health, College of Health Sciences, Addis Ababa University, Addis Ababa, Ethiopia, 3 Department of Psychiatry, School of Medicine, College of Health Sciences, Addis Ababa University, Addis Ababa, Ethiopia, 4 Department of Pediatric and Child Health Department, School of Medicine, College of Health Sciences, Addis Ababa University, Addis Ababa, Ethiopia, 5 Department of Surgery, School of Medicine, College of Health Sciences, Addis Ababa University, Addis Ababa, Ethiopia, 6 Department of Internal Medicine, School of Medicine, College of Health Sciences, Addis Ababa University, Addis Ababa, Ethiopia, 7 Department of Obstetrics and Gynecology, School of Medicine, College of Health Sciences, Addis Ababa University, Addis Ababa, Ethiopia, 8 Institute of Medical Epidemiology, Biostatistics and Informatics, Martin-Luther-University, Halle, Germany

* zelalem.desalegn@aau.edu.et, tzollove@gmail.com

**Data Availability Statement:** All relevant data are within the manuscript and its Supporting information files.

## Abstract

### Background

The COVID-19 pandemic is impacting the global community in many ways. Combating the COVID-19 pandemic requires a coordinated effort through engaging public and service providers in preventive measures. The government of Ethiopia had already announced prevention guidelines for the public. However, there is a scarcity of evidence-based data on the public knowledge, attitude, and practice (KAP) and response of the service providers regarding COVID-19.

### Objective

This study aimed to assess the public KAP and service providers' preparedness towards the pandemic in Addis Ababa, Ethiopia.

### Methods

A community-based cross-sectional study was conducted in Addis Ababa, Ethiopia, from late March to the first week of April 2020. Participants were conveniently sampled from 10 different city sites. Data collection was performed using a self-administered questionnaire and observational assessment using a checklist. All statistical analysis was performed using SPSS version Descriptive statistics, correlation coefficient and chi-square tests were performed.

**Funding:** ZD Grant number: VPRTT/PY-403/2020 Addis Ababa University www.aau.edu.et The funders have no role in in the study design, data collection and analysis, decision to publish, or preparation of the manuscript.

**Competing interests:** The authors have declared that no competing interests exist.

## Result

A total of 839 public participants and 420 service providers enrolled in the study. The mean age was 30.30 (range = 18–72) years. The majority of the respondents (58.6%) had moderate knowledge about COVID-19, whereas 37.2% had good knowledge. Moreover, 60.7% and 59.8% of the participants had a positive attitude towards preventive measures and good practice to mitigate the pandemic, respectively. There was a moderate positive correlation between knowledge and attitude, whereas the correlations between knowledge and practice and attitude and practice were weak. With regard to service providers' preparedness, 70% have made hand-washing facilities available. A large majority of the respondents (84.4%) were using government-owned media followed by social media (46.0%) as a main source of information.

## Conclusion

The public in Addis Ababa had moderate knowledge, an optimistic attitude and descent practice. The information flow from government and social media seemed successful seeing the majority of the respondents identifying preventive measures, signs and symptoms and transmission route of SARS-CoV-2. Knowledge and attitude was not associated with practice, thus, additional innovative strategies for practice changes are needed. Two thirds of the service provider made available hand washing facilities which seems a first positive step. However, periodic evaluation of the public KAP and assessment of service providers' preparedness is mandatory to combat the pandemic effectively.

## Introduction

Infections with coronaviruses in humans and animals cause respiratory and intestinal diseases [1]. The diseases vary from mild, self-limiting forms to more severe manifestations depending on the type of viruses involved [2]. Coronaviruses belong to the subfamily *Coronaviridae*, which consists of four genera: *Alphacoronavirus* and *Betacoronavirus* members infect mammals, while *Gammacoronavirus* and *Deltacoronavirus* only infect birds and some mammals [3]. Among the coronaviruses that infect humans, severe acute respiratory syndrome coronavirus (SARS-CoV) and middle East respiratory syndrome-related Coronavirus (MERS-CoV) are highly pathogenic [4].

The current human coronavirus, named SARS-CoV-2, emerged as a public health problem from Wuhan, Hubei province, China, on 31 December 2019 as a cluster of pneumonia cases. On 7 January 2020, the a etiological agent of the pneumonia was officially announced as a novel coronavirus [5–7]. On 11[th] January 2020, the first fatal case was reported. On the next day (12 January 2020), the whole genome sequence of the virus was shared with the World Health Organization (WHO) and the public. Confirmed cases outside Wuhan were reported from Thailand (13 January 2020), Japan (16 January 2020), Korea and in another province of China (19 January 2020), all from persons who had travelled to Wuhan [8]. On 30 January 2020, the Director-General of WHO declared the 2019-nCoV outbreaks a public health emergency of international concern [9]. The WHO announced that COVID-19 should be characterized as a pandemic on 11 March 2020 [9].

As of September 29, 2020, approximately 33,556,252 million cases, 1,006,450 deaths and 24,881,239 recovered cases have been reported globally [10]. Europe and America have been highly affected by the virus, as shown by overwhelmed health systems and high death tolls

[11]. Although the virus arrived late in Africa, the number is increasing and it has been predicted that more than 1.2 billion people are at high risk [12]. In the context of Ethiopia, the first COVID-19 case was reported on 13 March 2020. Based on WHO recommendations, Ethiopia implemented thermal screening at various institutions, social distancing, providing hand washing facilities, stay-at-home orders, quarantining people assumed to be exposed and encouraging the community to use homemade masks when needed, including in areas where there are more people and traffic flow such transportation services and other service providers. As of 29 September 2020, there had been 73, 944 confirmed cases, 1,177 deaths and 30, 753 recovered cases in Ethiopia [13].

According to the WHO global strategy to respond to COVID-19, the overarching goal of all countries is to control the pandemic by slowing down the transmission to reduce the immediate burden on health systems and to reduce the mortality [14]. According to this strategy, everyone has a crucial role to play to stop COVID-19. Individuals must protect themselves and others by adopting behaviors like regular adequate hand washing or use alcohol-based hand sanitizers, avoid touching their faces, practice covering their mouths and noses anytime or while coughing and sneezing, maintain physical distancing, isolate themselves if they are sick, identify themselves as a contact of confirmed cases when appropriate and, most importantly, strictly follow measures announced by their government or health authorities [14]. The implementation of all the above depends on the background knowledge, skills and attitude of the public to COVID-19.

The knowledge, attitudes and, practices (KAP) that people hold towards the disease play an integral role in determining a society's readiness to accept behavioral change measures from health authorities [15]. The KAP of people towards COVID-19 is critical to understand the epidemiological dynamics of the disease and the effectiveness, compliance and success of infection prevention control measures adopted in a country. Moreover, research has demonstrated that effective control and mitigation of COVID-19 in any country requires operational research and timely epidemiological data generated among different groups of the population. Such evidence-based data will inform health authorities so that they can design robust interventions and policies that are relevant and comprehendible to the community and beyond [16].

In a previous study, a plethora of evidence demonstrated that there is a disparity in the KAP level of the public about viral infection, including COVID-19 [15, 17–26]. The difference in the public KAP towards COVID-19 could be explained by geographical difference, methodological variability, health care system infrastructure, socio-economic status of the participants, the burden of the pandemic and residence of the participants, among many other factors.

The COVID-19 pandemic and the associated measures of confinement will have devastating consequences for micro and small business operations and will disrupt many existing value chains. This, in turn, will lead to loss of income and sharp increases in unemployment. The COVID 19 pandemic has and will continue to have a strong effect on labour markets worldwide, especially in developing economies, where more than 70% of the workforce is self-employed or works in micro and small enterprises [27, 28]. These effects will undeniably have many significant effects on a wide range of the population.

Engaging service providers and/or small and medium enterprises and exploring their preparedness to fight the COVID-19 pandemic is crucial. So far, government, health authorities, health institutions and the media have strived to help public and service providers be aware of the disease and apply preventive measures. Despite the public health measures, there is a huge research gap with regard to the public KAP and service providers' preparedness towards COVID-19.

Therefore, the present study aimed to assess: (1) the public KAP and (2) the preparedness and response of service providers towards COVID-19 in Addis Ababa, Ethiopia.

## Methods and materials

### Study design and setting

A cross-sectional, community-based survey was conducted in selected sites in Addis Ababa, Ethiopia, among adults to assess their KAP and the preparedness of the service providers regarding COVID-19. The study was conducted during the last week of March through first week of April 2020. To achieve the intended study objectives, a self-administered questionnaire was used to assess the public KAP and a brief checklist was utilized to evaluate service providers' preparedness and response towards COVID-19 in Addis Ababa, Ethiopia. Addis Ababa is the capital and largest city of Ethiopia. The study was carried out in 10 high traffic sites located in the respective sub-cities.

### Study population and the target sample size

The study population was adults who were by chance walking in the 10 sites. Service providers in the selected sites were considered for assessing their practical readiness against COVID-19. Being adult ($>$ 18 years of age) and consent to participate in the study taken were the inclusion criteria.

A single population proportion formula was used by considering 50% prevalence of public awareness of COVID-19, with a 5% margin of error at a 95% confidence level, with a design effect of 2.0, and adding 10% for non-response. A total of839(84/ per site) participants from major city sites were recruited. Again, given that there was no study among service providers related to a possible outbreak, we considered a 50% proportion of preparedness for COVID-19, with a 5% margin of error at a 95% confidence level, and adding 10% for non-response. Therefore, 420 service providers recruited from the 10 sites. A convenient sampling technique was employed and verbal consent obtained from all participants.

### Participant recruitment procedure

The intended study was conducted during the early phase of the COVID-19 pandemic in Ethiopia at Piazza, Arat Kilo, Mexico, Bole Medhanealem, Bole Michael, Teklehaimanot, Megenagna, Jemo, Ayer Tena and Kera. The selection of specific streets from the high traffic enumeration site was done by spinning a bottle. The participants were approached and informed of the study objectives. Consecutive service providers on the same streets were included; their preparedness was assessed using a brief checklist along with a direct observational assessment.

### Data collection tool

Data collection was done using a self-administered structured questionnaire and a brief checklist. The questionnaire consisted of 40 close-ended questions that aimed to collect the following information from the respondents: socio-demographic characteristics, travel history, risk factors and KAP related to COVID-19. The survey instrument took approximately 15–20 minutes to complete.

The data collection tool was initially prepared in English (S1 Appendix) version followed by translation to local Amharic language (S2 Appendix). Consistency of content, clarity and appropriate meaning between the two version was maintained through back translation of the questionnaire to the original version. Additionally, the practicability, validity and interpretability of answers for the respective questions was confirmed by performing pre-test in 10% of the targeted sample size. Based on this pre-test study, the format and wording of questions were corrected and refined.

A brief checklist and observational assessment were used to evaluate the preparedness and response of service providers (e.g. hotels, cafeterias and transportation enterprises). The brief checklist explored the availability of soap with water, alcohol and/or sanitizer for the any person entering. To facilitate the data collection, 10 data collection facilitators were enrolled to distribute and collect the completed questionnaire from the consented participants. Formal training on a brief introduction of the research objectives, data collection procedure and questionnaire content was delivered.

## Knowledge related to COVID-19

The knowledge section of the questionnaire comprised40 questions. All the questions were developed by considering previous research done in same population with a similar research theme [29]. These questions were in the form of yes or no; if the answer was yes, the participants were asked to specify. The right answer to each question has a score of 1 and wrong answer 0. Modified Bloom's cut-off points were used to judge knowledge as good (80%–100%),≥32), moderate (50%–79%, 20–31), or poor (≤ 50%,≤19) [30].

## Attitude towards COVID-19

Eight questions were asked to evaluate the attitude of the general public towards the disease. A scoring system to attitude was used as follows: good (≥6), moderate (5) and poor (≤4).

## Practice regarding COVID-19

There were four questions on practice towards COVID 19 (one point for each questions with correct answer). The cut-offs were similar to the knowledge and attitude scoring: good (4), moderate (3) and poor (≤2).

## Statistical analysis

Before the analysis, completeness of the data was evaluated. Data entry and coding and were done using EpiData version 3.1. The data were analyzed with SPSS statistical software version 22. A descriptive analysis was performed. Specific knowledge, attitude and practice questions were used to establish scoring to assess the overall status of the participants. For each question, 1 point was given for answering correctly, whereas 0 points were assigned when the responders fail to respond correctly. Based on the total score relative to the maximum score, the public KAP level was categorized as good, moderate or poor, considering modified Bloom's cut-off points. Inferential statistics between the socio-demographic factors and the public KAP regarding COVID-19 were investigated using a chi-square test. A statistically significant association was declared at $< 0.05$.

## Research ethics approval

The study protocol was approved by the institutional review board (IRB) of the College of Health Sciences Addis Ababa University (Protocol number: 012/DMIP/2020) and verbal consent was obtained from each participant.

# Results

## Demographic characteristics

The study included 839 participants. The participants mean age was 30.3(standard deviation [SD] = 9.25, range = 18–72) years. The majority of the respondents were males (58.0%) and

single (56.6%). With regard to occupational status, government employee and non-government employee occupied one third each (36.7% and 34.7%) followed by traders (8.3%), day workers (6.4%) and others (12.3%).

## Travel history to COVID-19-affected areas

Only 7% of respondents had travel history in the last three months at the time of data collection. Of these, 17%, 11.8% and 10.0% had travelled to China, Europe and the Middle East, respectively, among COVID-19-affected areas at the time of data collection. With respect to contact history, 9.2% of the participants had had contact with individuals who had travelled to COVID affected areas (Table 1).

## KAP towards the COVID-19 pandemic

**Knowledge assessment.** Knowledge was assessed using a total of 40 questions that focused on nature of the disease, prevention mechanisms, transmission mode, risk groups and signs and symptoms of COVID-19 (Table 2).

**Table 1. Demographic characteristics of the study participants in Addis Ababa, Ethiopia.**

| Characteristics | | Count (%) |
|---|---|---|
| **Sex** | Male | 487 (58.0) |
| | Female | 345 (41.1) |
| **Age group (years)** | ≤19 | 48 (5.7) |
| | 20–29 | 426 (50.8) |
| | 30–39 | 233 (27.8) |
| | 40–49 | 82 (9.8) |
| | 50–59 | 31(2.7) |
| | ≥60 | 10 (1.2) |
| | Unknown age | ? |
| **Marital status** | Single | 475(56.6) |
| | Married | 322 (38.4) |
| | Divorced | 31 (3.7) |
| | Widowed | 7 (0.8) |
| | Unknown | ? |
| **Occupation** | Governmental | 308 (36.7) |
| | Non-governmental | 291(34.7) |
| | Trader | 70 (8.3) |
| | Day worker | 54 (6.4) |
| | Others | 103 (12.3) |
| | Unknown | ? |
| **Living condition** | Alone | 191 (22.8) |
| | With others | 629 (75.0) |
| | Unknown | ? |
| **Travel history** | Yes | 59 (7.1) |
| | No | 777(92.9) |
| **Contact with a person who travelled to COVID-19 affected areas** | Yes | 69 (9.2) |
| | No | 677(90.6) |
| | Unknown | |

**Table 2. The response of the participants to specific knowledge questions in Addis Ababa, Ethiopia.**

| Ser. No. | Knowledge questions | Responses | Correct response | Wrong response |
|---|---|---|---|---|
| 1 | Which of the following do you think are the major signs and symptoms of the disease caused by coronavirus? | 1. Fever | 721 (85.9) | 118 (14.1) |
| | | 2. Diarrhea | 68 (8.1) | 771 (91.9) |
| | | 3. Bloody diarrhea | 816 (97.3) | 23 (2.7) |
| | | 4. Bloody sputum | 786 (93.7) | 53 (6.3) |
| | | 5. Swelling of legs | 822 (96.0) | 17 (2.0) |
| | | 6. Cough | 412 (49.1) | 427 (50.9) |
| | | 7. Swelling on mouth/nose | 790 (94.2) | 49 (5.8) |
| | | 8. Red and painful eyes | 29 (3.5) | 810 (96.5) |
| | | 9. Sneezing/runny nose | 532 (63.4) | 307 (36.6) |
| 2 | What are the current ways of prevention of COVID-19? | 1. Vaccination | 728 (86.8) | 111 (13.2) |
| | | 2. Anti-viral therapy | 775 (92.4) | 64 (7.6) |
| | | 3. Using masks | 387 (46.1) | 452 (53.9) |
| | | 4. Frequent washing of hands | 446 (53.2) | 393 (46.8) |
| | | 5. Staying at home | 622 (74.1) | 217 (25.9) |
| | | 6. Frequent disinfectant | 504 (60.1) | 335 (39.9) |
| | | 7. Staying >meters from others | 542 (64.6) | 297 (35.4) |
| 3 | How could a person acquire the coronavirus disease? | 1. Directly through breathing/ sneezing | 698 (83.2) | 141 (16.8) |
| | | 2. Through a mosquito bite | 757 (90.2) | 82 (9.8) |
| | | 3. Touching mouth and nose through contaminated hand | 657 (78.3) | 182 (21.7) |
| | | 4. Through unprotected sexual intercourse | 121 (14.4) | 121 (14.4) |
| | | 5. Through staying and playing near others | 169 (20.1) | 670 (79.9) |
| | | 6. Not frequently washing while at work | 327 (39.0) | 512 (61.0) |
| | | 7. Using public transport with closed windows | 448 (53.4) | 391 (46.6) |
| | | 8. Opening doors/windows in public places | 477 (56.9) | 362 (43.1) |
| | | 9. Frequent use of disinfectant while at work | 774 (92.3) | 65 (7.7) |
| 4 | Who is at risk of developing a severe form of the corona disease? | 1. Diabetic patients | 531 (63.3) | 308 (36.7) |
| | | 2. Hypertensive patients | 403 (48.0) | 436 (52.00 |
| | | 3. People with heart problem | 449 (53.5) | 390 (46.5) |
| | | 4. Pregnant women | 555 (66.2) | 284 (33.8) |
| | | 5. Cancer patients | 379 (45.2) | 460 (54.8) |
| | | 6. Khat chewers/smokers | 432 (51.5) | 407 (48.5) |
| | | 7. Asthmatic patients | 440 (52.4) | 399 (47.6) |
| | | 8. People with COPD | 627 (74.7) | 212 (25.3) |
| 5 | At what age group do you think the coronavirus disease occur? | 1. Children | 413 (49.2) | 426 (50.8) |
| | | 2. Youth | 485 (57.8) | 354 (42.2) |
| | | 3. Elderly | 760 (90.6) | 79 (9.4) |
| 6 | Is the coronavirus transmittable by shaking/hugging anyone? | | 777 (92.6) | 39 (4.6) |
| 7 | Is coronavirus transmittable by mosquito bite? | | 588 (70.1) | 242 (28.8) |
| 8 | Is the coronavirus transmittable by direct breathing? | | 694 (82.7) | 133 (15.9) |
| 9 | Is a person who has coronavirus detectable by looking at him/ her? | | 713 (85.0) | 120 (14.3) |

The majority of respondents (58.6%) had moderate knowledge while37.2% had good knowledge (Table 3). Among the socio-demographic characteristics, only the age and occupation of the participants was associated with knowledge (Table 4).

**Attitude towards COVID-19 and association with demographic characteristics.** A total of eight questions were used to assess the attitude of the participants to implement preventive measures against the COVID-19 pandemic. As shown in Table 3, the mean attitude score was 5.73, most of the public had positive attitude (60.7%) towards implementation of preventive

**Table 3. Number of questions, range, scores and levels of knowledge, attitude and practice of the study participants in Addis Ababa, Ethiopia.**

| Variables | Number of questions | Score range | Total score mean ± SD | Level (points) | | |
|---|---|---|---|---|---|---|
| | | | | Poor | Moderate | Good |
| Knowledge | 40 | 16–40 | 28.92±5.4 | 35 (4.2) | 492(58.6) | 312 (37.2) |
| Attitude | 8 | 0–8 | 5.73±2.1 | 247 (29.4) | 82 (9.8) | 509 (60.7) |
| Practice | 4 | 0–4 | 2.49±0.7 | 94 (11.2) | 242 (28.8) | 502 (59.8) |

*Note.* SD, standard deviation.

**Table 4. Association between respondent demographic characteristics and level of knowledge, attitude and practice scores in Addis Ababa, Ethiopia.**

| Characteristics | | Knowledge scores | | | P | Attitude scores | | | p | Practice scores | | | p |
|---|---|---|---|---|---|---|---|---|---|---|---|---|---|
| | | Poor | Mod. | Good | | Poor | Mod. | Good | | Poor | Mod. | Good | |
| | | N (%) | N (%) | N (%) | | N (%) | N (%) | N (%) | | N (%) | N (%) | N (%) | |
| Sex | Male | 24 (4.9) | 292 (60.0) | 171 (35.1) | 0.09 | 145 (29.8) | 49 (10.1) | 293(60.2) | 0.87 | 61 (12.5) | 140 (28.7) | 286 (58.7) | 0.41 |
| | Female | 9 (2.6) | 196 (56.8) | 140 (40.6) | | 98 (28.5) | 33 (9.6) | 213 (61.9) | | 33 (9.6) | 100 (29.1) | 211 (61.3) | |
| Age group (years) | ≤ 19 | 5 (10.4) | 33 (68.8) | 10 (20.8) | 0.06 | 19 (39.6) | 6 (12.5) | 23 (47.9) | 0.58 | 5 (10.4) | 17 (35.4) | 26 (54.2) | 0.63 |
| | 20–29 | 18 (4.2) | 250 (58.7) | 158 (37.1) | | 125 (29.3) | 36 (8.45) | 265 (62.2) | | 47 (11.0) | 124(29.1) | 255 (59.9) | |
| | 30–39 | 4 (1.7) | 131 (56.2) | 98 (42.1) | | 69 (29.7) | 22 (9.5) | 141 (60.8) | | 25 (10.8) | 63 (27.2) | 144 (62.1) | |
| | 40–49 | 3 (3.7) | 50 (61.0) | 29 (35.4) | | 19 (23.2) | 11 (13.4) | 52 (63.4) | | 8 (9.8) | 26 (31.7) | 48 (58.5) | |
| | 50–59 | 3 (9.7) | 17 (54.8) | 11(35.5) | | 9 (29.0) | 5 (16.1) | 17 (54.8) | | 6 (19.3) | 9 (29.0) | 16 (51.6) | |
| | ≥ 60 | 1 (10.4) | 6 (60.0) | 3 (30.0) | | 2 (20.0) | 1 (10.0) | 7(70.0) | | 0 (0) | 1 (10.0) | 9 (90.0) | |
| Marital Status | Un-married | 18 (3.8) | 277 (58.3) | 180 (37.9) | 0.16 | 131 (27.6) | 48 (10.1) | 295 (62.2) | 0.40 | 46 (9.7) | 147 (31.0) | 281 (59.3) | <0.05 |
| | Married | 11 (3.4) | 194 (60.2) | 117(36.3) | | 98 (30.4) | 28 (8.7) | 196 (60.9) | | 42 (13.0) | 82 (25.5) | 198 (61.5) | |
| | Divorced | 4 (12.9) | 15 (48.4) | 12 (38.7) | | 14 (45.2) | 4 (12.9) | 13 (41.9) | | 2 (6.5) | 11(35.5) | 18 (58.1) | |
| | Widowed | 1 (14.3) | 4 (57.1) | 2 (28.6) | | 2 (28.6) | 1(14.3) | 4 (57.1) | | 3 (42.9) | 2(28.6) | 2 (28.6) | |
| Occupation | Govern-mental | 9 (2.9) | 158 (51.3) | 141 (45.8) | <0.05 | 82 (26.7) | 18 (5.9) | 207 (67.4) | <0.05 | 29 (9.4) | 89 (29.0) | 189 (61.6) | 0.67 |
| | Non-govern-mental | 17 (5.8) | 182 (62.5) | 92 (31.6) | | 96 (33.0) | 39 (13.4) | 156 (53.6) | | 38 (13.1) | 80 (27.5) | 173 (59.5) | |
| | Trader | 2 (2.9) | 42 (60.0) | 26 (37.1) | | 20 (28.6) | 6 (8.6) | 44 (62.9) | | 8 (11.4) | 18 (25.7) | 4 (62.9) | |
| | Day worker | 2 (3.7) | 41 (75.9) | 11 (20.4) | | 17 (31.5) | 7 (13.0) | 30 (55.6) | | 8 (14.8) | 20 (37.0) | 26 (48.1) | |
| | Others | 3 (2.9) | 59 (57.3) | 41 (39.8) | | 25 (24.3) | 10 (9.7) | 68 (66.0) | | 10 (9.7) | 30 (29.1) | 63 (61.2) | |

measures against COVID-19. Among the respondents, 83.1% and 74.9% indicated they prefer frequent hand washing with soap and water and use alcohol-based sanitizer, respectively. Moreover, the majority (90.3%) had good attitude towards social distancing and its necessity to prevent COVID-19. With regard to lockdown, more than half of the participants agreed that it had to be in place to mitigate the pandemic in Ethiopia. Similar to knowledge, only occupational status was associated with a positive attitude (Table 4).

**Practice towards COVID-19.** In the study, there were four questions related to practice towards COVID-19, with a maximum total of four points. The mean practice score was 2.49 ± 0.7(range 0–4). The majority (59.8%) of the study participants had a good practice towards COVID-19. On the date of the data collection, the study participants' experience of hand washing with soap and water for 20 seconds and utilization of sanitizer was 96.4% and 82.2%, respectively. Similarly, 88.0% of the participants had not practiced hand shaking. Good practice was only associated with marital status (Table 4).

**Correlations among knowledge, attitude and practice.** To visualize the correlation of participants knowledge, attitude and practice with one another, we performed a scatter plot analysis. There was a moderate positive correlation between participant's knowledge and attitude (r = 0.624), whereas the correlations between knowledge and practice (r = 0.196) and attitude and practice (r = 0.172) were weak (Table 5).

## Source of information about COVID-19

For the majority of the respondents (84.4%), government-owned television was the primary source of information about COVID-19, followed by government-owned radio (49.7%), social media (46.0%) and private television (43.0%). Besides, the government health and social media were the information sources that the respondents highly believed. With regard to the adequacy of information, more than half (59.6%) of the respondents believed that the broadcasted information was adequate to act against COVID-19.

## Response of service providers towards COVID-19 pandemic

The checklist used to assess the service providers contained questions on type of enterprise, method of preventive mechanism in place and the type of washing facility present (Table 6).

Most (70%) of the enterprises provided hand washing facilities as a preventive mechanism towards COVID-19, followed by social distancing and sanitizer or alcohol use with 8.6% and 7.9%, respectively (Table 7).

Of the enterprises, 264 (62.9%) had hand washing facilities with soap and water, 32(7.6%) had water only, 11 (2.6%) had soap only and 34 (8.1%) had none of the hand washing facilities (Table 8).

## Discussion

This study is the first survey in the capital of Ethiopia, Addis Ababa as far as our knowledge goes, that aimed to assess the public KAP towards the COVID-19 pandemic as well as to assess the preparedness and response of service providers in the city.

**Table 5. Correlation between knowledge, attitude and practice scores towards COVID-19.**

|   | Variables | Correlation coefficient | P |
|---|---|---|---|
| A | Knowledge and attitude | 0.624 | <0.01 |
| B | Knowledge and practice | 0.196 | <0.01 |
| C | Attitude and practice | 0.172 | <0.01 |

**Table 6. Type of service providers included in the study in Addis Ababa, Ethiopia.**

|   | Enterprise type | Number | Percent |
|---|---|---|---|
| 1 | Hotel/restaurant/cafeteria/juice house | 114 | 27.1 |
| 2 | Bus/taxi/train station | 26 | 6.2 |
| 3 | Banks | 69 | 16.4 |
| 4 | Local drinking houses | 21 | 5.0 |
| 5 | Mall/boutiques, cosmetic shops, business centre, stationary | 85 | 20.2 |
| 6 | Others | 103 | 24.5 |

*Note*. Others include electronics shops, butchers, pharmacies, bakeries, churches, mosques, etc.

**Table 7. Preventive measures made available by service providers in Addis Ababa, Ethiopia.**

|   | Preventive measure | Number | Percent |
|---|---|---|---|
| 1 | Hand washing facility (soap and water) | 294 | 70 |
| 2 | Sanitizer/alcohol | 33 | 7.9 |
| 3 | Social /physical distancing | 36 | 8.6 |
| 4 | None | 97 | 23.1 |
| 5 | Both hand washing facility and sanitizer/alcohol | 13 | 3.1 |
| 6 | Both hand washing facility and social/physical distancing | 28 | 6.7 |
| 7 | Both sanitizer/alcohol and social physical distancing | 7 | 1.7 |

**Table 8. Type of washing facility available to prevent COVID-19 among service providers in Addis Ababa, Ethiopia.**

| Facility | Number | Percent |
|---|---|---|
| **Water only** | 32 | 7.6 |
| **Soap only** | 11 | 2.6 |
| **Both (water and soap)** | 264 | 62.9 |
| **None** | 34 | 8.1 |

In the survey out of 839 respondents, almost two thirds had moderate knowledge and good attitude and practice. This level was far lower than a multinational survey in Africa (South Africa, Kenya and Nigeria), which reported that the level of awareness and concern about COVID-19 were very high (94%) [26]. A bi-national survey in Egypt and Nigeria also demonstrated that the mean knowledge score was higher, with a satisfactory knowledge of the disease [22]. A study from Nigeria also proved that the respondents had good knowledge (99.5%) of COVID-19 [16]. Since the current study in Addis Ababa was carried out during the early phase of the pandemic, the reported knowledge level is encouraging; however, periodic assessment should be in place considering the different scenario of COVID-19 pandemic in Ethiopian setting.

A study on Indian diabetes mellitus populations reported a high overall correct response rate on the knowledge questionnaire (83%) [31]. In another study, the majority of the participants were knowledgeable about COVID-19, with a mean COVID-19 knowledge score of 17.96 (SD = 2.24; Range = 3–22), indicating a high level of knowledge and the overall accuracy rate for the knowledge test was 81.64% (17.96/22 _ 100) [21]. A high knowledge level has also been reported in Malaysia, where the overall correct rate of the knowledge questionnaire was 80.5%and most participants held positive attitudes towards the successful control of COVID-19 (83.1%) [15].

The burden of COVID-19 was by far higher in some Asian countries than some African countries including Ethiopia, such difference in the spread would bring a disparity in the overall knowledge status of the population and preparedness towards the pandemic. Though the current knowledge and preparedness status is descent in our setting, the best practice from other countries employed to improving knowledge and preparedness should be adapted for best containment of the pandemic.

Knowledge assessment in this study included signs and symptoms, the disease transmission mode, the prevention mechanisms and risk groups. According to the assessment, a considerable number of the participants were aware of the disease signs and symptoms. However, a few participants incorrectly attributed signs and symptoms not shown in COVID-19 cases. This finding is similar to a study from the Philippines; those results showed that coughing and sneezing were identified as a transmission route by 89.5% of respondents [23]. In our study, knowledge of fever and cough as COVID-19 symptoms was high, and the participants knew that younger participants had a lower perceived risk and the elderly were identified as the high risk group [32]. One study from the United States among people with chronic conditions provided unexpected results: nearly one third could not correctly identify symptoms (28.3%) or ways to prevent infection (30.2%) [25].

Very interestingly, during early phase of COVID-19 pandemic, there has been an aggressive promotion of covid19 information through MOH and main government mass media. This lead to better knowledge and preparedness about the pandemic. Though still the promotion is present, adherence seems to become less. We believe that preventive attitude has to be re-enforced and appropriate prevention and control strategies should be promoted consistently.

With respect to identifying knowledge question related to COVID-19 prevention, nearly 50% of the participants identified using face mask, frequent hand washing and staying at home as the most important means of preventing the pandemic. The finding was by far lower than a study from Philippines which showed that hand washing was the most common preventive practice in response to COVID-19, adopted by 89.9% of respondents [23]. Another report from Ethiopia demonstrated that even 90% of the participants had a good prevention knowledge of maintaining social distance and frequent hand washing [24]. The moderate knowledge in our survey of participants living in the capital city of Ethiopia with consistent access to information.

Our study explored the association of socio-demographic characteristics with the public KAP. There was only an association between occupational status and good knowledge. In contrast with our findings, study from Tanzania and Iran showed that male sex, younger age (16 to 29 years), non-healthcare-related professions, being single and less education were significantly associated with lower knowledge scores [32, 33].

We also assessed the attitude of the participants towards practicing preventive measures, perceptions on lockdown and their stand on staying at home. Concerning attitudes, it was interesting that close to two thirds of the respondents showed a positive and optimistic attitude towards COVID-19 preventive measures. Similarly, a study from Saudi Arabia demonstrated that the mean score for attitude indicated optimistic attitudes and the mean score for practices was high, indicating good practices [21]. Findings from Egypt and Nigeria indicated that the attitude of most respondents (68.9%) towards instituted preventive measures was positive, with an average attitude score of 6.9 ± 1.2. In addition, the majority of the respondents (96%) practiced self-isolation and social distancing [22].

Another finding among the same population from Africa documented that the majority of the respondents (79.5%) had positive attitudes towards adherence to government infection prevention and control (IPC) measures, with 92.7%, 96.4% and 82.3% practicing social distancing/self-isolation, improved personal hygiene and using face masks, respectively [16].

In agreement with participants knowledge, the state of their attitude towards applying the preventive measures has been positive. Moreover, the findings proved that those with moderate knowledge and good knowledge turned out to have positive attitude which could ultimately impact the practice of the public and response towards for any possible outbreak.

The aforementioned optimistic attitude was consistent with participants' practice of washing hands with soap and water and frequent use of hand sanitizer. It is an established fact that physical distancing is the most effective but also the most challenging measure. The respondents had a positive attitude towards physical distancing and implementation of lockdown in Ethiopia. This positive attitude will ultimately help in the prevention and control of COVID-19. However, periodic evaluation of this positive attitude towards preventive measures must be performed to determine whether it is sustained among the public.

In support of the present findings, a study from Ethiopia among several population revealed that frequent hand washing (77.3%) and avoiding shaking hands (53.8%) were the dominant practices [34]. Unlike our study, another investigation among health professionals from Oromia regional state, Ethiopia reportedly demonstrated that the practices of the participants towards COVID-19 prevention were relatively low: only 61% and 84% of the participants were practicing social distance and frequent hand washing, respectively [24]. Such a discrepancy might be due to the difference in the study population, study area and the pandemic phase.

In our study, only knowledge and attitude showed a moderate correlation. A previous study showed stronger relationship between knowledge, attitude and practice with infection prevention measures [35]. Finding from China revealed that COVID-19 knowledge score (odds ratio [OR] 0.75–0.90, p<0.001) was significantly associated with a lower likelihood of negative attitudes and preventive practices towards COVID-2019 [17]. This finding were also reported from Malaysia where most participants held positive attitudes towards the successful control of COVID-19 (83.1%) [15].

During an emergency, timely, adequate and appropriate information is important as the best intervention against rumors and misinformation [5]. Following the emergence of the pandemic, a large amount of information has been released in media based on internet information about COVID-19. Based on previous assessment, only 1.9% websites that provide health-related information had agreed to the Net Foundation Code of Conduct by the time of assessment [36].

The study explored the source of information regarding COVID-19. The majority of the study participants (84.4%) obtained information from government-owned television broadcast, followed by government-owned radio broadcast, social media and private television broadcast. In line with our finding, study from Iran indicated that government TV advertisements and short message service (SMS) were the most common sources of COVID-19 information and considered trustworthy (by >95% of participants) [32]. This was in support of a research finding from Philippines which demonstrated that traditional media sources such as television and radio were the main sources of information about the virus [23]. By contrast, another recent study in Ethiopia reported that social media were the main source of the information [24].

It was interesting that the public source of information was government outlets at the early phase of the pandemic; however, with time the public also tended to use social media as the primary source of information [24, 26]. Another study from Nigeria found that the participants mainly gained information about COVID-19 through the internet/social media (55.7%) and television (27.5%) [16]. However, the quality of information shared on the social media requires due attention and regulation to provide the public with reliable information so as to combat the pandemic effectively and in a sustainable approach.

TheCOVID-19 pandemic has been affecting enterprises of all sizes and types in unprecedented way [27, 37]. The majority of the assessed service providers in Addis Ababa in April 2020 had made available either washing facilities with soap and water or alcohol-based hand rub in an accessible spot. The availability of the washing facilities might explain the moderate state of knowledge, good attitude and best practice of public KAP. This is a very encouraging response; it shows that the government strategies were acceptable to the public, stake holders and clients of the service providers.

This survey had some limitations. First, the convenience sampling method did not avoid subjective selection bias. Second, selected localities may not reflect the whole picture of Addis Ababa at large because the ten sites were selected purposefully considering high traffic flow. In addition, we used a descriptive cross-sectional study design, which hinders determining a cause–effect relationship between an independent variable and the outcome variables. The comparability to other studies may be limited by use of different questionnaires. Although the study faced the above mentioned limitations, the strength of this study lies in its large sample size. To our knowledge, this is the first large scale study considering the public and service providers KAP and preparedness towards COVID-19 pandemic.

In terms of policy implication, the findings will the policy makers reconsider the engagement of the community as a key approach in combating any possible outbreak, including COVID 19. In general, data from the current study showed most probably a positive public health education effect leading to desired preventive measures as recommended by the government in the city.

## Conclusion

In conclusion, the finding suggested that the public in Addis Ababa had moderate knowledge, optimistic attitudes and notable practice against the COVID-19 pandemic. Government and social media seem valuable sources of information and should further be utilized. COVID-19 knowledge correlated with an optimisticattitudetowardsCOVID-19; these finding indicate that effective awareness creation and health education have been delivered.

The service providers' level of preparedness towards the pandemic was encouraging. Still, more practical support seems needed to assure full coverage with hand hygiene options in public enterprises. Periodic evaluation of service providers awareness and preparedness for any possible outbreak should be in place to assure sustainability of efforts.

## Supporting information

**S1 Appendix.**
(PDF)

**S2 Appendix.**
(PDF)

## Acknowledgments

We would like to express our gratitude to health professionals and researchers working to overcome COVID-19 throughout the world during this critical time.

## Author Contributions

**Conceptualization:** Zelalem Desalegn, Negussie Deyessa, Damen Hailemariam, Adamu Addissie, Tamrat Abebe.

**Data curation:** Zelalem Desalegn, Negussie Deyessa, Brhanu Teka, Welelta Shiferaw, Damen Hailemariam, Adamu Addissie, Abdulnasir Abagero, Mirgissa Kaba, Workeabeba Abebe, Berhanu Nega, Wondimu Ayele, Tewodros Haile, Yirgu Gebrehiwot, Wondwossen Amogne, Eva Johanna Kantelhardt, Tamrat Abebe.

**Formal analysis:** Zelalem Desalegn, Brhanu Teka, Welelta Shiferaw, Adamu Addissie, Abdulnasir Abagero, Mirgissa Kaba, Workeabeba Abebe, Wondimu Ayele, Eva Johanna Kantelhardt, Tamrat Abebe.

**Funding acquisition:** Zelalem Desalegn, Negussie Deyessa, Tamrat Abebe.

**Investigation:** Zelalem Desalegn, Negussie Deyessa, Brhanu Teka, Welelta Shiferaw, Damen Hailemariam, Adamu Addissie, Abdulnasir Abagero, Mirgissa Kaba, Workeabeba Abebe, Berhanu Nega, Wondimu Ayele, Tewodros Haile, Yirgu Gebrehiwot, Wondwossen Amogne, Eva Johanna Kantelhardt, Tamrat Abebe.

**Methodology:** Zelalem Desalegn, Negussie Deyessa, Welelta Shiferaw, Adamu Addissie, Abdulnasir Abagero, Mirgissa Kaba, Workeabeba Abebe, Berhanu Nega, Wondimu Ayele, Wondwossen Amogne, Eva Johanna Kantelhardt, Tamrat Abebe.

**Project administration:** Zelalem Desalegn, Brhanu Teka, Welelta Shiferaw, Damen Hailemariam, Abdulnasir Abagero, Mirgissa Kaba, Berhanu Nega, Wondwossen Amogne, Eva Johanna Kantelhardt, Tamrat Abebe.

**Resources:** Zelalem Desalegn.

**Supervision:** Zelalem Desalegn, Brhanu Teka, Welelta Shiferaw, Adamu Addissie, Tewodros Haile, Yirgu Gebrehiwot, Wondwossen Amogne, Eva Johanna Kantelhardt, Tamrat Abebe.

**Validation:** Zelalem Desalegn, Negussie Deyessa, Brhanu Teka, Welelta Shiferaw, Damen Hailemariam, Adamu Addissie, Abdulnasir Abagero, Mirgissa Kaba, Workeabeba Abebe, Berhanu Nega, Wondimu Ayele, Tewodros Haile, Yirgu Gebrehiwot, Wondwossen Amogne, Eva Johanna Kantelhardt, Tamrat Abebe.

**Visualization:** Zelalem Desalegn, Brhanu Teka, Welelta Shiferaw, Damen Hailemariam, Adamu Addissie, Abdulnasir Abagero, Mirgissa Kaba, Workeabeba Abebe, Berhanu Nega, Wondimu Ayele, Tewodros Haile, Yirgu Gebrehiwot, Wondwossen Amogne, Eva Johanna Kantelhardt, Tamrat Abebe.

**Writing – original draft:** Zelalem Desalegn, Brhanu Teka, Welelta Shiferaw, Abdulnasir Abagero, Wondimu Ayele, Tamrat Abebe.

**Writing – review & editing:** Zelalem Desalegn, Negussie Deyessa, Brhanu Teka, Welelta Shiferaw, Damen Hailemariam, Adamu Addissie, Abdulnasir Abagero, Mirgissa Kaba, Workeabeba Abebe, Berhanu Nega, Wondimu Ayele, Tewodros Haile, Yirgu Gebrehiwot, Wondwossen Amogne, Eva Johanna Kantelhardt, Tamrat Abebe.

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
