## [Decision Letter · Decision Letter 0]

2 Jul 2020

PONE-D-20-17538

COVID-19 and the public response: knowledge, attitude and practice of the public in mitigating the pandemic in Addis Ababa, Ethiopia

PLOS ONE

Dear Dr. Desalegn,

Thank you for submitting your manuscript to PLOS ONE. After careful consideration, we feel that it has merit but does not fully meet PLOS ONE’s publication criteria as it currently stands. Therefore, we invite you to submit a revised version of the manuscript that addresses the points raised during the review process.Please submit your revised manuscript by Aug 16 2020 11:59PM. If you will need more time than this to complete your revisions, please reply to this message or contact the journal office at plosone@plos.org. Please include the following items when submitting your revised manuscript:

We look forward to receiving your revised manuscript.

Kind regards,

Khin Thet Wai, MBBS, MPH, MA (Population & Family Planning Resear

Academic Editor

PLOS ONE

Additional Editor Comments:

This manuscript highlights the KAP gaps focusing the social service sector which are mostly used by the general public. For further strengthening of research , authors should consider the following in addition to reviewers' comments.

1. Extensive English language editing is deemed necessary.

2. Authors need to discuss the limitations of the study and policy implications.

3. Authors need to add one descriptive table analyzing 40 knowledge items.

Journal Requirements:

5. Please amend the manuscript submission data (via Edit Submission) to include author Wondimu Ayele.

6. Please amend your list of authors on the manuscript to ensure that each author is linked to an affiliation. Authors’ affiliations should reflect the institution where the work was done (if authors moved subsequently, you can also list the new affiliation stating “current affiliation:….” as necessary).

7. We note you have included a table to which you do not refer in the text of your manuscript. Please ensure that you refer to Table 6 in your text; if accepted, production will need this reference to link the reader to the Table.

Reviewers' comments:

Reviewer's Responses to Questions

**Comments to the Author**

1. Is the manuscript technically sound, and do the data support the conclusions?

Reviewer #1: Partly

Reviewer #2: Yes

Reviewer #3: Partly

2. Has the statistical analysis been performed appropriately and rigorously? 

Reviewer #1: Yes

Reviewer #2: Yes

Reviewer #3: No

3. Have the authors made all data underlying the findings in their manuscript fully available?

Reviewer #1: No

Reviewer #2: No

Reviewer #3: Yes

4. Is the manuscript presented in an intelligible fashion and written in standard English?

Reviewer #1: No

Reviewer #2: Yes

Reviewer #3: No

5. Review Comments to the Author

Reviewer #1: PONE-D-20-17538

COVID-19 and the public response: knowledge, attitude and practice of the public in mitigating the pandemic in Addis Ababa, Ethiopia

The manuscript addresses the important topic in timely presentation by describing the KAP of community in mitigating COVID-19 pandemic. While the manuscript is of some interest and paucity of COVID-19 related data, the manuscript could be strengthened by several modest changes as outlined below.

I suggest the authors to look for STROBE check list for cross-sectional studies to ensure reporting is complete and transparent.

https://www.strobe-statement.org/fileadmin/Strobe/uploads/checklists/STROBE_checklist_v4_cross-sectional.pdf

GENERAL COMMENTS

-The manuscript is not well-written and needs to be edited by a native English speaker. Please check typo, grammar mistakes and format throughout the manuscript.

-Inserting line numbers may facilitate to give comments and feedbacks.

ABSTRACT

-The sentence “This would need knowledge, attitude and practice (KAP) of the population” is unclear.

-The authors concluded “The public service providers and enterprises were well prepared to contribute in the measures against the diseases”. But presented 62.9% made hand washing facilities available which is not satisfactory especially for COVID-19 prevention.

-Results presented here must be best support to the understanding of conclusions.

INTRODUCTION

-The sentence “The diseases vary from mild, self-limiting diseases to more severe manifestations depending on the type of viruses involved” is not clear. What types of virus involved? Subtypes or genotypes? Are you mentioning different types of viruses or COVID only?

-The sentence “The current human coronavirus named SARS-CoV-2 emerged as a public health problem from Wuhan City, Hubei Province of China on 31 December 2019 as a cluster of pneumonia cases” needs reference.

-In the sentence “As of June 7 , there were 2020 confirmed cases, twenty seven and 344 recovered cases in Ethiopia”, what do you mean by “twenty seven and 344 recovered cases”? typo error?

-The introduction section becomes like the history of COVID-19 but Why prevention plays vital role for COVID-19? Why do you need to assess KAP? are missing.

METHODS

-Did you calculate sample size for service providers?

-Is fever screening include in the observation checklist? If not, why?

-The authors mentioned a total of 35 closed questions including socio-demographic characteristics, travel history, risk factors, and KAP, and then later mentioned 40 Knowledge questions, 8 attitude questions and 4 practice questions.

-Add reference for Bloom’s cut off point

-Data analysis should be elaborated more

RESULTS

-The results section should avoid discussion words like interestingly, unlike, etc. and be written in academic way (E.g. “Next we looked at the association of knowledge…..”).

-Table 1 – check and correct frequency and percentage.

-Could you add travel and contact history to Table 1.

-I could not find contact history in the results. Do you ask question like “have contact with COVID-19 positive patient?”? as the authors titled travel and contact history.

-The description regarding variables included in the questionnaires should be moved to Methods.

-Did you calculate knowledge level by specific knowledge themes: prevention, transmission, sign and symptoms, etc. ?

-Any reason for using correlation coefficient (r) among KAP but not chi-square (as authors used before)?

-Table 7 – can the enterprise types combined into categories? (E.g. Hotel/restaurant and Cafeteria) as 35% occupied as others.

-I do not find the result in Table 8 for the sentence “Of the enterprises 264 (62.9%) had hand-washing facilities with soap and water, 32(7.6%) of them had water only, 11 (2.6%) of them had soap only and 34 (8.1%) had neither of the washing facilities (table 8)”. Meanwhile, hand washing facility of 294 (70%) from the Table 8 has discrepancy with the above mentioned sentence. Is hand washing facility 294 or 264 or 264+34 or 264+32+11? What is your operational definition for hand washing facility?

-The authors mentioned as small and medium enterprises but I found bank and mall in the list. Is bank and mall SME in Ethiopia?

DISCUSSION

-The discussion is weak in light of the findings and should be rewritten. The discussion needs to focus on the key implications of the data with a separate paragraph for each concept and discuss the potential reasons for it by comparing local and international literatures. Moreover, repeating the finding statements and analytical term (E.g. p value) should be avoided here.

-Some discussed points have not presented in the results (E.g. Nearly two thirds of the respondents could not properly identify symptoms or know how the disease is transmitted and could not identify preventive measures).

Reviewer #2: This manuscript describes the results of a community based KAP survey in Ethiopia

Some comments

1- The abbreviations, SARS CoV, MERS needs to be defined

2- In the abstract, its described that the Questionnaire was self administered ; in the M&M, it was administered by data collectors. Authors should clarify on this

3. In the introduction, the number of cases, recoveries and deaths needs to be clarified

4. The questionnaire needs to be provided as a supplementary material for the readers

5. Discussion, some sentences missing citations

Reviewer #3: The authors attempt to describe the knowledge, attitude and practices about COVID-19 prevention and mitigation practices among members of the community, service providers and enterprises in Addis Ababa, Ethiopia.

The study requires a major overhaul of the language for better comprehension

Abstract

The line in conclusion “The public service providers and enterprises were well prepared to contribute in the measures against the diseases.” Does not stem from the study findings which are only about hand washing facilities’ availability.

Main text

The penultimate para in introduction mentions twenty seven….but does not qualify what it is referring to

The introduction does not establish the rationale for studying the preparedness of the service providers

There is no sample size calculation described for the enterprises and service providers

Why a design effect of 2 was chosen is not described

The authors have calculated the sample size for descriptive analysis / proportion..but the analysis plans included comparison of two proportions..

They have written “The modified Bloom’s cutoff points were used to judge knowledge as good, moderate or poor if the total mark is :sufficient knowledge ≥80%, positive attitude :80-100% (≥32) good;50-79% (20- 31) moderate ;o r ≤ 50% (≤19) poor knowledge respectively.” It is not clear if it is for knowledge or attitude

It says data was validated, but not elaborated how? Was double data entry done for validation?

The nature of consent obtained isn’t clear

From the data presented in tables, the questionnaire seems to be very arbitrarily designed.

Table 3,4,5 can be compressed into one

The larger picture of the relevance of the findings and their addition to existing knowledge to inform current prevention and control measures for COVID

6. PLOS authors have the option to publish the peer review history of their article (what does this mean?). If published, this will include your full peer review and any attached files.

Reviewer #1: **Yes: **Kyaw Lwin Show

Reviewer #2: No

Reviewer #3: No

---

## [Author Response · Author response to Decision Letter 0]

29 Sep 2020

Point by point response for Editor and reviewers comment

Manuscript number: PONE-D-20-17538 

Manuscript title: "COVID-19 and the public response: knowledge, attitude and practice of the public in mitigating the pandemic in Addis Ababa, Ethiopia"

Dear editor and the reviewer, 

First of all, we would like to thank the journal team members, the journal editor and the respective reviewers for taking their time and sending valuable suggestions for our manuscript. 

As per the request, we incorporated all the relevant changes to the revised manuscript with track change. Additionally, a detail point by point response for each editors and reviewers comment enclosed with this letter.

With kind regards, 

Zelalem Desalegn Woldesonbet

PI and corresponding/submitting author

Assistant professor, Department of Microbiology, Immunology and Parasitology

School of Medicine, College of Health Sciences

Addis Ababa University

Reviewer 1 comments:

The manuscript addresses the important topic in timely presentation by describing the KAP of community in mitigating COVID-19 pandemic. While the manuscript is of some interest and paucity of COVID-19 related data, the manuscript could be strengthened by several modest changes as outlined below. 

Reviewer 1-General comments

1. Comments: Consider STROBE check list for cross-sectional studies to ensure reporting is complete and transparent

Dear reviewers, we appreciate your suggestion and direction. We have gone through the STROBE for cross-sectional studies, and forwarded thoughtful consideration into the method section.

2. Comments: Please check typo, grammar mistakes and format throughout the manuscript

The authors and a professional proof-reading service thoroughly checked and corrected the manuscript. 

3. Comments: Inserting line numbers may facilitate to give comments and feedbacks.

We accept the raised suggestion. The line numbers are included in the revised version of the manuscript.

4. Comments: the manuscript is not well-written and needs to be edited by a native English speaker

We considered the comment. To manage the language issue, professional proof-reading was implemented.

Reviewer 1-Specific comments: Abstract

1. Comments: The sentence “This would need knowledge, attitude and practice (KAP) of the population” is unclear. 

Yes, we accept your thoughtful comment. Accordingly, we have presented the unclear aforementioned sentence in a very meaningful way in the revised version of the manuscript (“However, there is a scarcity of evidence-based data on the public knowledge, attitude and practice ( KAP) and response of the service providers regarding COVID-19. Therefore, the study was conducted to generate key findings attributable to combating the recent pandemic ”)(Please refer line number 40 to 43)

2. Comments: The authors concluded “The public service providers and enterprises were well prepared to contribute in the measures against the diseases”. But presented 62.9% made hand washing facilities available which is not satisfactory especially for COVID-19 prevention.

We agree with your argument. Definitely, only making available the hand washing facility in 62.9% may be too little to have significant impact on COVID 19 prevention, but we consider this as a good starting point. Therefore, the phrase presented is now: “Two thirds of public service providers made hand washing facilities available is a first positive step”(please refer line number 69 to 70).

3. Comments: Results presented here must be best support to the understanding of conclusions.

Based on the suggestion made, the result section has been re-constructed taking into account additional relevant findings and the conclusion was drawn out of it. In addition to the previously presented data, we made additions of the major findings which was missed due to the word limit of the abstract. 

Reviewer 1-Specific comments: Introduction

1. Comments: The sentence “The diseases vary from mild, self-limiting diseases to more severe manifestations depending on the type of viruses involved” is not clear. What types of virus involved? Subtypes or genotypes? Are you mentioning different types of viruses or COVID only?

The sentence "“The diseases vary from mild, self-limiting diseases to more severe manifestations depending on the type of viruses involved” in the first paragraph is a continuation of the first sentence referring " Infections with Coronaviruses in humans and animals cause respiratory and intestinal diseases " describing about infection caused by any type of Coronavirus not specifically with the SARS-cov-2.

2. Comments: The sentence “The current human coronavirus named SARS-CoV-2 emerged as a public health problem from Wuhan City, Hubei Province of China on 31 December 2019 as a cluster of pneumonia cases” needs reference

Thank you. The indicated research finding along with other related information was extracted from a references 5 through 7 which has been also indicated in the submitted manuscript and in the reference section( please refer line number 92).

3. Comments: In the sentence “As of June 7, there were 2020 confirmed cases, twenty seven and 344recovered cases in Ethiopia”, what do you mean by “twenty seven and 344recovered cases”? typo error?

Well taken. As you have noted it well, in the description we missed to link the figure" twenty seven" with the health parameter which is "deaths" during the submission. We have made changes of the numbers taking into account the latest data( Please refer line number 101 to 102).

4. Comments: The introduction section becomes like the history of COVID-19 but Why prevention plays vital role for COVID-19? Why do you need to assess KAP? are missing.

Well taken. Evidence based data showing the relevance of the study and the reason for the finding on KAP are presented well and the role of prevention to fight against the pandemic well narrated concisely with latest data released on COVID-19. “The KAP of people towards COVID-19 is critical to understand the epidemiological dynamics of the disease and the effectiveness, compliance and success of infection prevention control measures adopted in a country. Moreover, research has demonstrated that effective control and mitigation of COVID-19 in any country requires operational research and timely epidemiological data generated among different groups of the population ( Please refer line number 125 to 153).

Reviewer 1-Specific comments: Methods

1.Comments: Did you calculate sample size for service providers? 

Well taken. originally, we calculated the sample size but missed in the previously submitted manuscript. Considering your comments, the sample size calculation assumptions with the final sample size presented under the method section in the revised version of the manuscript ( please refer line number 176 to 179).

2. Comments : Is fever screening include in the observation checklist? If not, why?

We did not include fever screening in the checklist because at that time point the government and health authorities only advised the service providers to make available washing facilities or alternatives including sanitizer. Additionally, the enterprises were enforced to manage the sitting pattern of the customer. In case of for example transportation service, the users were well spaced while standing in line. In case of restaurants and Bank, security officers directed the customers to wash their hands before entry for the respective services. We did not include it into the checklist because thermal screening was not in place during the early phase of the pandemic in out context. 

3. Comments :The authors mentioned a total of 35 closed questions including socio-demographic characteristics, travel history, risk factors, and KAP, and then later mentioned 40 Knowledge questions, 8 attitude questions and 4 practice questions.

Yes, you are right and well taken. We changed to 40 knowledge questions( please refer line number 193).

4. Comments :Add reference for Bloom’s cut off point

Well taken. Reference incorporated in the revised version of the method section( line number 202). 

5. Comments :Data analysis should be elaborated more

We did the elaboration on data analysis and incorporated into the revised version of the manuscript. (second last paragraph of method section)( please refer 224 to 233)

Reviewer 1-Specific comments: Results

1. Comments: The results section should avoid discussion words like interestingly, unlike, etc. and be written in academic way(E.g. “Next we looked at the association of knowledge…..”). 

Thank you for the forwarded comments. Based on your comments, we omit of using words highlighting description of matter. 

2. Comments : Table 1 – check and correct frequency and percentage.

Thank you. In table 1, there were few variables to which the participants did not respond for example in case of age and others. The proportion of unknown information was added.

3. Comments :Could you add travel and contact history to Table 1.

Well taken. As per the suggestion, the travel and contact history is added to Table 1.

4. Comments I could not find contact history in the results. Do you ask question like “have contact with COVID-19 positive patient?”?as the authors titled travel and contact history.

Well taken. We did not include whether the participants had contact with COVID-19 patients instead we asked the history of contact with a person who travelled to COVID affected areas. This is changed.

5. Comments: The description regarding variables included in the questionnaires should be moved to Methods.

We have considered your comments and moved to method section

6. Comments: Did you calculate knowledge level by specific knowledge themes: prevention, transmission, sign and symptoms, etc. ?

Well taken. We have created a new descriptive table summarizing presented knowledge questions focusing on prevention, sign and symptoms, transmission. Additionally, there are too few questions to further sub-classify the knowledge and a further study on this could be interesting. 

7. Comments :Any reason for using correlation coefficient (r) among KAP but not chi-square (as authors used before)?

We have used correlation coefficient because it shows the direction of the relation between Knowledge-attitude, Knowledge-practice, Attitude-practice in addition to Chi-square which shows only presence of association across/characteristics. 

In addition, we have seen that other research findings similar study population applied correlation coefficient and thus comparison is possible. 

Comments: 7 – can the enterprise types combined into categories? (E.g. Hotel/restaurant and Cafeteria) as 35% occupied as others.

Yes, you are perfectly correct. During the analysis we summarized the enterprise by putting them into in same package However, the table number was re-organized the required table was labeled as table 6 in the revised version(Please refer line number starting from 321)

8. Comments : I do not find the result in Table 8 for the sentence “Of the enterprises 264 (62.9%) had hand-washing facilities with soap and water, 32(7.6%) of them had water only, 11 (2.6%) of them had soap only and 34 (8.1%) had neither of the washing facilities (table 8)”. Meanwhile, hand washing facility of 294 (70%) from the Table 8 has discrepancy with the above mentioned sentence. 

Just to be clear, 264 (62.9%) had hand-washing facilities with soap and water is referring to the detail of the facility, whereas the hand washing facility of 294 (70%) was referring hand washing facility regardless of the type of the hand washing facilities made available by the service providers.

For the better understanding of these part, we have added additional table describing the type of facilities in each service providers (Please refer table 7 and 8). 

9. Comments : Is hand washing facility 294 or 264 or 264+34 or 264+32+11? What is your operational definition for hand washing facility?

Well taken and we appreciate your perspectives. When we are saying hand washing facility, a facility packaged with water with soap. To make the figure more clear, we have created additional table( please refer table 8)

10. The authors mentioned as small and medium enterprises but I found bank and mall in the list. Is bank and mall SME in Ethiopia?

Thank you for your observation. We have considered your suggestion into the revised version of the manuscript. “A total of 420 service providers were included in the survey.”

Reviewer 1- Specific comments: Discussion

1. Comments The discussion is weak in light of the findings and should be rewritten. The discussion needs to focus on the key implications of the data with a separate paragraph for each concept and discuss the potential reasons for it by comparing local and international literatures. Moreover, repeating the finding statements and analytical term (E.g. p value) should be avoided here.

We have considered your comments and the discussion was re-written considering the major findings of the study. We attempted to provide strong evidence-based information conducted elsewhere and tried to explain our finding, appreciate the discrepancy and strengthen our studies importance with respect to showing the gaps in the fight against the pandemic. 

Additionally, Considering your suggestions, we avoided using the detailed findings in the discussion section. 

2. Comments: Some discussed points have not presented in the results (E.g. Nearly two thirds of the respondents could not properly identify symptoms or know how the disease is transmitted and could not identify preventive measures). 

Based on your scientific comments, all the discussed points have been incorporated into the result section. Your comments were valuable and accommodated accordingly.

Reviewer 2 comments

Reviewer #2: This manuscript describes the results of a community based KAP survey in Ethiopia

1. Comments: The abbreviations, SARS CoV, MERS needs to be defined

This has been addressed in the revised manuscript (please refer line numbers 86 and 87).

2. Comments: In the abstract, its described that the Questionnaire was self administered ; in the M&M, it was administered by data collectors. Authors should clarify on this

Well noted. Yes, as you have mentioned the questionnaire was self administered; however, the data collectors were responsible for facilitating, briefing the objectives of the study, distributing the tool, checking the completeness and collecting the questionnaire from each consented participants ( please refer line number 192 and 207).

See Methods: “To facilitate the data collection, 10 data collection facilitators were enrolled to distribute and collect the completed questionnaire from the consented participants. Formal training included a brief introduction of the research objectives, data collection procedure and questionnaire content was delivered.”( please refer line number 207 to 210)

3. Comments : In the introduction, the number of cases, recoveries and deaths needs to be clarified

We highly appreciate your comments. Accordingly, we presented the local and global latest update of SARS cov-2 number of cases, recoveries and death figure with the respective reference. Please see introduction.

Global perspectives: "As of September 29 2020, approximately 33,556,252 million cases, 1,006,450 deaths and 24,881,239 recovered cases have been reported globally" ( please refer line number 101 and 102)

Ethiopian context: "As of 29 September 2020, there had been 73, 944 confirmed cases, 1,177 deaths and 30, 753 recovered cases in Ethiopia" ( line number 112 and 113)

4. Comments: The questionnaire needs to be provided as a supplementary material for the readers

Well taken comments. The questionnaire provided as a supplementary material in the revised version of the manuscript. The detail description of the questionnaire presented under the method section( please refer line number 192 to 210). 

5. Comments : Discussion, some sentences missing citations

We thoroughly looked at the manuscript to make sure that there are sentences missing citations. However, the citations are sometimes mentioned few sentences later in the text.

Reviewer #3: 

The authors attempt to describe the knowledge, attitude and practices about COVID-19 prevention and mitigation practices among members of the community, service providers and enterprises in Addis Ababa, Ethiopia.

1. Comments :The study requires a major overhaul of the language for better comprehension

Well taken. A major language editing was done throughout the manuscript. The revised version has also been sent out for professional language editing service. 

2. Comments : The line in conclusion “The public service providers and enterprises were well prepared to contribute in the measures against

the diseases.” Does not stem from the study findings which are only about hand washing facilities’ availability.

Well taken. As you have clearly described, we have modified the sentences in the conclusion part of the abstract section. (“Two thirds of public service providers made hand washing facilities available is a first positive step.”)( Please refer line number 69 and 70)

3. Comments : The penultimate para in introduction mentions twenty seven….but does not qualify what it is referring to

Well noted. 

In the introduction section, twenty seven was referring death related with COVID 19 in Ethiopian context. In the revised manuscript, latest data related with COVID 19 has been presented in the local context(please refer line number 112 and 113).

4. Comments : The introduction does not establish the rationale for studying the preparedness of the service providers

Well taken. 

Considering your comments, we have presented research based information and established facts describing how service providers preparedness could affect the prevention and control of any possible outbreak including COVID 19( please refer line number 141 to 153)

5. Comments :There is no sample size calculation described for the enterprises and service providers. Why a design effect of 2 was chosen is not described

Sample size calculation was done for the service providers and shown in the revised version of the manuscript ( please refer line number 176 to 179). The design effect was used to maximize the sample size and enhance the generalization of the finding.

6. Comments : The authors have calculated the sample size for descriptive analysis / proportion..but the analysis plans included comparison of two proportions.

Well taken. We carried out both descriptive analysis and association of the independent variables with the outcome variables computed using Chi-square – this was rather exploratory analysis.

7. Comments: They have written “The modified Bloom’s cutoff points were used to judge knowledge as good, moderate or poor if the total

mark is :sufficient knowledge ≥80%, positive attitude :80-100% (≥32) good;50-79% (20- 31) moderate ;o r ≤ 50% (≤19)poor knowledge respectively.” It is not clear if it is for knowledge or attitude .

Thank you for your insightful comments. The description was meant for knowledge not for attitude. The clarification presented very well in the revised version of the manuscript. The above description ws replaced with " The right answer to each question has a score of 1 and wrong answer 0. Modified Bloom’s cut-off points were used to judge knowledge as good (80%–100%), ≥32), moderate (50%–79%, 20–31),or poor (≤ 50%, ≤19) "( please refer line number 216 to 218) . Moreover, the modified bloom's cut-off point was referenced in the method section of the revised version of the manuscript.

8. Comments : It says data was validated, but not elaborated how? Was double data entry done for validation?

Well taken. No double entry was done was not done and thus not described due to lack of resources. 

9. Comments: The nature of consent obtained isn’t clear

Verbal consent was obtained from each participants which has been indicated under participant recruitment procedure and ethical approval section( please refer line number 239) .

10. Comments : From the data presented in tables, the questionnaire seems to be very arbitrarily designed.

We appreciate your comments. What we did was, we explored research works conducted in related topic and same population. Following that, we referred also WHO and CDC guidelines, to select major themes and outline the content which have to be considered while assessing KAP of the public towards COVID 19. Finally, we have built the questions taking into account the local social, cultural and educational context. 

11. Comments : Table 3,4,5 can be compressed into one

The larger picture of the relevance of the findings and their addition to existing knowledge to inform current prevention and control measures for COVID

Well taken, and table 3, 4 and 5 merged and a new table was created ( table 4) 

Additional Editor Comments:

This manuscript highlights the KAP gaps focusing the social service sector which are mostly used by the general public. For further strengthening of research , authors should consider the following in addition to reviewers' comments.

1. Extensive English language editing is deemed necessary.

Professional English language editing was done.

2. Authors need to discuss the limitations of the study and policy implications.

Well taken. We have considered the need of discussing the limitation and anticipated policy implication of the finding. Accordingly, the comment incorporated into the revised version of the manuscript (discussion section)( please refer line number 517 to 530). 

3. Authors need to add one descriptive table analyzing 40 knowledge items.

Well noted. Accordingly, we have presented a descriptive table composed of knowledge questions on prevention, sign and symptom and transmission mechanisms of SARS-cov-2( Please refer table 2).

---

## [Decision Letter · Decision Letter 1]

15 Oct 2020

PONE-D-20-17538R1

COVID-19 and the public response: knowledge, attitude and practice of the public in mitigating the pandemic in Addis Ababa, Ethiopia

PLOS ONE

Dear Dr. Desalegn,

Thank you for submitting your manuscript to PLOS ONE. After careful consideration, we feel that it has merit but does not fully meet PLOS ONE’s publication criteria as it currently stands. Therefore, we invite you to submit a revised version of the manuscript that addresses the points raised during the review process.Please submit your revised manuscript by Nov 29 2020 11:59PM. If you will need more time than this to complete your revisions, please reply to this message or contact the journal office at plosone@plos.org. Please include the following items when submitting your revised manuscript:

We look forward to receiving your revised manuscript.

Kind regards,

Khin Thet Wai, MBBS, MPH, MA (Population & Family Planning Resear

Academic Editor

PLOS ONE

Additional Editor Comments (if provided):

Still needs to do English language editing by the native speaker or the recognized English language editing service and also needs to improve the discussion part up to standard.

Reviewers' comments:

Reviewer's Responses to Questions

**Comments to the Author**

1. If the authors have adequately addressed your comments raised in a previous round of review and you feel that this manuscript is now acceptable for publication, you may indicate that here to bypass the “Comments to the Author” section, enter your conflict of interest statement in the “Confidential to Editor” section, and submit your "Accept" recommendation.

Reviewer #1: (No Response)

Reviewer #2: All comments have been addressed

2. Is the manuscript technically sound, and do the data support the conclusions?

Reviewer #1: Yes

Reviewer #2: Yes

3. Has the statistical analysis been performed appropriately and rigorously? 

Reviewer #1: Yes

Reviewer #2: Yes

4. Have the authors made all data underlying the findings in their manuscript fully available?

Reviewer #1: Yes

Reviewer #2: Yes

5. Is the manuscript presented in an intelligible fashion and written in standard English?

Reviewer #1: Yes

Reviewer #2: Yes

6. Review Comments to the Author

Reviewer #1: PONE-D-20-17538-R1

COVID-19 and the public response: knowledge, attitude and practice of the public in mitigating the pandemic in Addis Ababa, Ethiopia

The authors have made their efforts and the manuscript becomes far better than before. However, the manuscript still could be strengthened especially in the discussion.

-LINE 166 – I think it is a typo. “839 84 per site)”.

-LINE 232 – 36.7 % is not the large majority. You can simply say” government employee and non-government employee occupied one third each (36.7% and 34.7%) followed by …”.

-Table 1 - Remove the unknown row if it does not exit (0%) or remain if occupied some %

-Some parts of table 4 are missing. I think it is a formatting error.

-Table 5 - I accept author explanation. It is better to present as a correlation matrix table.

-Table 8 - does not make 100% in cumulative. Please check.

-Discussion – still have lots of room for improvement for better, concise and comprehensive. Repeating the finding statements and sentences from the methods (e.g. LINE 341, 369, 377, etc.) should be avoided here. Information is repeated in many places (e.g. LINE 329 vs 360 vs 402). There are ways to discuss better without repeating findings and methods. The discussion part includes mostly comparison while missing potential reasons why the current results were found.

Reviewer #2: I would like to thank the authors for sufficiently addressing all the comments raised by the reviewer. Well done.

7. PLOS authors have the option to publish the peer review history of their article (what does this mean?). If published, this will include your full peer review and any attached files.

Reviewer #1: **Yes: **Kyaw Lwin Show

Reviewer #2: **Yes: **Felix Bongomin, MD

---

## [Author Response · Author response to Decision Letter 1]

15 Dec 2020

Response to reviewers and editor comments:

Reviewer #1: 

COVID-19 and the public response: knowledge, attitude and practice of the public in mitigating the pandemic in Addis Ababa, Ethiopia

The authors have made their efforts and the manuscript becomes far better than before. However, the manuscript still could be strengthened especially in the discussion.

1. LINE 166 – I think it is a typo. “839 84 per site)”.

Thank you for your thoughtful comment. We appreciate the comment and corrected in the revised version ( Please refer line number 166 of the revised version)

2. LINE 232 – 36.7 % is not the large majority. You can simply say” government employee and non-government employee occupied one third each (36.7% and 34.7%) followed by …”.

Well taken. We accept the comment and we presented the data in the way you have recommended ( Please refer line number 232-236 of the revised manuscript)

2. Table 1 - Remove the unknown row if it does not exit (0%) or remain if occupied some %

Well taken. We have removed the unknown row ( Please refer table 1)

3. Some parts of table 4 are missing. I think it is a formatting error.

Well taken. We included the missed p-value ( Please refer table 4)

4. Table 5 - I accept author explanation. It is better to present as a correlation matrix table.

We appreciate your positive feedback.

5. Table 8 - does not make 100% in cumulative. Please check.

Well taken. Yes, as you have sated it has to give 100%; however, the reason for not making 100% is due to no response for some of specific questions in the checklist ( please refer table 8)

6. Discussion – still have lots of room for improvement for better, concise and comprehensive. Repeating the finding statements and sentences from the methods (e.g. LINE 341, 369, 377, etc.) should be avoided here. Information is repeated in many places (e.g. LINE 329 vs 360 vs 402). There are ways to discuss better without repeating findings and methods. The discussion part includes mostly comparison while missing potential reasons why the current results were found.

We appreciate your comments. Accordingly, we removed the finding statements and sentence directly taken from result section. We also removed information repeated and tried to discuss better without repeating findings and methods (Please refer the whole discussion part for better clarification).

Reviewer #2:

 I would like to thank the authors for sufficiently addressing all the comments raised by the reviewer. Well done.

Additional Editor Comments (if provided): 

1. Still needs to do English language editing by the native speaker or the recognized English language editing service and also needs to improve the discussion part up to standard.

Well take. We appreciate the comment forwarded by editor for the betterment of the revised manuscript. We have gone through the manuscript thoroughly to improve the language and present the research data fulfilling the journal requirements. 

Additionally, the manuscript has been sent out for recognized language editing service. 

The authors strived to their best to make the discussion part very comprehensive, concise and clear to improve it up to the standard.

---

## [Editor Report · Decision Letter 2]

17 Dec 2020

COVID-19 and the public response: knowledge, attitude and practice of the public in mitigating the pandemic in Addis Ababa, Ethiopia

PONE-D-20-17538R2

Dear Dr. Desalegn,

We’re pleased to inform you that your manuscript has been judged scientifically suitable for publication and will be formally accepted for publication once it meets all outstanding technical requirements.

Kind regards,

Khin Thet Wai, MBBS, MPH, MA (Population & Family Planning Res.)

Academic Editor

PLOS ONE
---

## [Editor Report · Acceptance letter]

28 Dec 2020

PONE-D-20-17538R2 

COVID-19 and the public response: knowledge, attitude and practice of the public in mitigating the pandemic in Addis Ababa, Ethiopia 

Dear Dr. Desalegn:

I'm pleased to inform you that your manuscript has been deemed suitable for publication in PLOS ONE. Congratulations! Your manuscript is now with our production department. 

Kind regards, 

on behalf of

Dr. Khin Thet Wai 

Academic Editor

PLOS ONE